# Molecular Subtypes as a Basis for Stratified Use of Neoadjuvant Chemotherapy for Muscle-Invasive Bladder Cancer—A Narrative Review

**DOI:** 10.3390/cancers14071692

**Published:** 2022-03-26

**Authors:** Gottfrid Sjödahl, Johan Abrahamsson, Carina Bernardo, Pontus Eriksson, Mattias Höglund, Fredrik Liedberg

**Affiliations:** 1Department of Translational Medicine, Lund University, Malmö and Department of Urology Skåne University Hospital, Jan Waldenströms gata 5, 21421 Malmö, Sweden; 2Division of Oncology, Department of Clinical Sciences, Lund University, Scheelevägen 2, 22381 Lund, Sweden

**Keywords:** urothelial carcinoma, bladder cancer, neoadjuvant, cisplatin, chemotherapy, response, biomarker, molecular subtypes, luminal, basal

## Abstract

**Simple Summary:**

Although it is one disease, cancer of the urinary bladder occurs in several molecular subtypes that can be identified by laboratory tests. Tumors of advanced stages are treated with surgical removal of the urinary bladder with or without addition of chemotherapy. About 50% of patients are cured by surgery and this proportion is increased slightly by the addition of chemotherapy. Still, many patients do not benefit from chemotherapy, which also comes with significant toxicity. Recent advances in the field suggest that molecular subtypes can help identify patient categories that do or do not benefit from adding chemotherapy to surgery. In this article, we review the literature and conclude that molecular subtypes are likely to have such a role in the future but that there are differences between studies that make them challenging to compare. The current evidence is insufficient to guide clinical practice.

**Abstract:**

There are no established biomarkers to guide patient selection for neoadjuvant chemotherapy prior to radical cystectomy for muscle-invasive bladder cancer. Recent studies suggest that molecular subtype classification holds promise for predicting chemotherapy response and/or survival benefit in this setting. Here, we summarize and discuss the scientific literature examining transcriptomic or panel-based molecular subtyping applied to neoadjuvant chemotherapy-treated patient cohorts. We find that there is not sufficient evidence to conclude that the basal subtype of muscle-invasive bladder cancer responds well to chemotherapy, since only a minority of studies support this conclusion. More evidence indicates that luminal-like subtypes may have the most improved outcomes after neoadjuvant chemotherapy. There are also conflicting data concerning the association between biopsy stromal content and response. Subtypes indicative of high stromal infiltration responded well in some studies and poorly in others. Uncertainties when interpreting the current literature include a lack of reporting both response and survival outcomes and the inherent risk of bias in retrospective study designs. Taken together, available studies suggest a role for molecular subtyping in stratifying patients for receiving neoadjuvant chemotherapy. The precise classification system that best captures such a predictive effect, and the exact subtypes for which other treatment options are more beneficial remains to be established, preferably in prospective studies.

## 1. Introduction

Cisplatin-based neoadjuvant chemotherapy (NAC) followed by radical cystectomy (RC) is the current standard of care for patients with muscle-invasive bladder cancer (MIBC) and its use has increased over the last decade [1]. Adding systemic immunotherapy to cisplatin combinations in the neoadjuvant setting seems to further improve the proportion of patients with downstaging in the cystectomy specimen based on phase II data [2,3]. In addition to the current use in patients ineligible for cisplatin-based chemotherapy, checkpoint inhibitors are also likely to become relevant in the neoadjuvant setting as single agent or in combination with chemotherapy [4]. Furthermore, bladder-sparing chemoradiotherapy is an alternative curative treatment for selected patients [5]. Currently, no molecular markers are used to guide neoadjuvant treatment with chemo- or immunotherapy. Suggested molecular biomarkers in the published literature can be broadly divided into four categories:Single or panels of genes/proteins expressed by the cancer cells;Markers and cell types present in the tumor microenvironment;Presence of genomic alterations, e.g., specific gene mutations in the cancer cells;Classification of tumors into molecular subtypes based on the whole transcriptome.

Since the early 1990s, many single or panels of expressed RNA/protein biomarkers have been suggested to be predictive of cisplatin responsiveness, usually to either of the two most common NAC regimens, M-VAC (metotrexate, vinblastine, Adriamycin, and cisplatin) or GC (gemcitabine and cisplatin). Among the more frequently studied expressed tissue biomarkers are proteins such as Bcl-2, p21, p53, SLC31A1 (CTR1), ABCB1 (MDR1, P-Glycoprotein), BRCA1, MRE11, ERCC1, NFE2L2 (Nrf2), and BSG (emmprin). This field has been reviewed elsewhere [6,7], and despite rational mechanisms for several of the proposed markers, none have been prospectively validated or shown to result in a patient selection that leads to improved treatment outcomes.

Several studies have also described potentially important links between tumor microenvironments, including density of tumor-infiltrating lymphocytes, or other inflammatory cells, and response to neoadjuvant treatment, either with immunotherapy [8,9] or NAC [10,11,12,13]. A comprehensive review of prognostic and predictive biomarkers present in the microenvironment has been published [14].

At the genomic level, the most studied biomarkers suggested to predict NAC response are specific mutations in DNA damage response and repair (DDR) genes. The prime example is the presence of hotspot mutations in nucleotide-excision repair gene *ERCC2*, which was initially identified in 9/25 complete responders but in 0 out of 25 non-responders [15]. This association has subsequently been validated in an independent cohort of similar size [16], and helicase domain mutations have been experimentally confirmed to cause cisplatin sensitivity [17]. In addition to *ERCC2*, inactivating mutations in *FANCC*, *RB1*, and *ATM* [18], *ERBB2* [19], and *BRCA2* [20], or mutations in a panel of 34 DDR-related genes [21], have been suggested to correlate with NAC response, but none of these associations have been independently replicated by other researchers.

In recent years, molecular subtype classification of MIBC has also gained interest as a potential biomarker for neoadjuvant treatment response. Currently, basic scientific studies have broadly established which molecular tumor categories exist, but it is not yet clear if and how such a classification provides clinically useful information. When molecular classification is to be translated into clinical use, it is important to consider that the several classification methods emphasize slightly different aspects of tumor biology. The classification systems differ in the details of how tumors are grouped into molecular categories based on the transcriptomic profiles (reviewed in [22]). A recent consensus publication showed that in terms of tumor biology, all existing classification systems converge on the same biological themes and are in general agreement with each other [23]. However, when it comes to translating molecular subtypes into clinical use, such a general agreement is not enough. In this new scenario, it is imperative that the basis for tumor stratification is precise and that the method for classification is specified in detail. It is a general problem that subtypes and classification methods are often only loosely defined, even when the study aim has shifted from biological description to clinical translation. For example, many studies define subtyping simply as an axis between luminal and basal categories, even though we know that the reality is more complex. Facing this methodological diversity, the consensus subtypes will play an important role as they serve as a ‘common ground’ that can be applied in addition to any other classifier to make translational results comparable between groups. Therefore, every translational effort that uses RNA-based subtyping of bladder cancer must also apply and share case-wise results with the consensus classification. Although RNA-based classification is considered the golden standard for subtyping, full transcriptomic analysis is still relatively costly and comes with requirements for the amount and quality of tissue samples. Therefore, there are relatively few NAC-treated cohorts with original transcriptomic data. A potentially more accessible way forward could be to apply immunostaining [24,25] or RNA-based qPCR or Nanostring panels [26,27] as a surrogate for whole transcriptome molecular subtype classification. The strategy to infer molecular subtypes based on limited data can extend to cohorts and case series that did not intend to classify tumors into molecular subtypes, if case-wise data are available for subtype-defining markers. Another important aspect of molecular classification is that it can be performed at different levels, resulting in a different number of subtypes [28]. The most appropriate level must be determined empirically for each classification system and translational question. The possibility to group tumors by several different classification systems and at different levels causes analytic flexibility. To protect the field from bias, it is therefore important that molecular subtyping studies use pre-specified classifiers and logical groupings based on, e.g., biological similarity or statistical power, rather than post hoc testing of several clustering solutions and subtype groupings that best separate the clinical data. Testing several versions of classifiers, e.g., different algorithm parameters, cut-offs, clustering solutions, or groupings of subtypes, can under some circumstances be a good course of action, but it must always be disclosed.

Based on the current literature, molecular subtyping of MIBC has been shown to be prognostic in cohorts with a wide variation in treatment and clinicopathological risk factors [23,29], but less so in more stringently selected cohorts [30,31]. An arguably more important translational question is if applying subtyping to tissue from trans-urethral resection of bladder tumor (TURB) adds purely treatment-predictive information independent of clinical and pathological factors known at the time of neoadjuvant treatment decision. If this proves to be the case, molecular subtyping could both enable effective neoadjuvant treatment where it is currently not routinely used, and/or limit overtreatment where it is currently universally applied. The number of publications on this topic, mostly retrospective cohort studies, is increasing rapidly. The aim of this review is to summarize the available evidence and provide an outlook for future research in the field.

## 2. Methods

This narrative review aims to identify, describe, and summarize available studies on molecular subtyping of MIBC and neoadjuvant treatment response. The review includes studies employing molecular subtype categorization of invasive bladder cancer through either:Original RNA-classifiers based on one of several existing molecular classification systems for bladder cancer derived from transcriptomic data;Markers or panels that in combination are used as a surrogate for an existing molecular classification system, e.g., in the form of an IHC, qPCR, or Nanostring panel;Markers or panels for cellular differentiation states (e.g., luminal-like, non-luminal, and basal-like) that are clearly related to molecular subtype categories. Such studies may be included even if there was no intention to divide the cohort into molecular subtypes if marker combinations matching known subtype categories can be deduced from the results.

We did not include studies that apply combinations of biomarkers not meeting these criteria, nor did we include molecular classifications based on (epi)genetic data. All the identified studies that fulfilled these criteria were included and discussed in the results section. We describe the study design, subtyping method, and main findings of each study, with an emphasis on specific aspects that are considered relevant for interpretation. The directionality and statistical significance of the associations are summarized in the tables, but no meta-analysis is performed because of the large variability in subtyping definitions. Studies on NAC with original mRNA subtyping data are summarized in Table 1.

## 3. Results

### 3.1. Studies on NAC Response with Original mRNA-Subtyping Data

The first study that compared an RNA-based classification to clinical endpoints in patients receiving NAC was published in 2014 by Choi and colleagues [32]. This study used expression data (HT-12 and DASL microarrays, Illumina, San Diego, CA, USA) from three cohorts (MDA-GC, MDA-MVAC, and the Philadelphia cohort) with a total of 100 NAC-treated patients. Each tumor was classified as ‘basal’, ‘p53-like’, or ‘luminal’ using a nearest-neighbor centroid classifier developed by the MD-Anderson Cancer Center (MDA) group. The composite response rates were MDA-basal 16/33 (48%), MDA-p53-like 3/30 (10%), and MDA-luminal 20/37 (54%). Response was defined as ypT0 or as ypT1 if the patient had high-risk features at TURB, including lympho-vascular infiltration, variant histology, hydronephrosis, or abnormal exam under anesthesia denoting cT3–4 disease. It is unclear if these high-risk criteria were pre-defined, and if pN0 was used as a response criterion.

In 2016, the same group published an analysis of response to neoadjuvant dose-dense MVAC plus bevacizumab in relation to the MDA classification [33]. The study included pretreatment gene expression profiling for 38 patients using DASL microarrays (Illumina). Classification was not associated with pathologic response. The number of patients in each subtype who achieved a pT0 were 4/11 MDA-basal, 3/11 MDA-Luminal, and 2/16 MDA-p53-like. In this study, pTis/pTa/pT1 were also considered responses and by these criteria the MDA-luminal subtype had the highest response rate (8/11, 73%) compared to MDA-basal (5/11, 45%) and MDA-p53-like (5/16, 31%). Regarding overall survival (OS), MDA-p53-like had worse outcomes compared to patients with tumors of the MDA-luminal and MDA-basal subtypes.

One year later, Seiler and colleagues published a multi-center study comparing NAC-response and survival in 251 patients with gene expression profiling data [34]. Several existing classification systems were applied, and a new genomic subtyping classifier (GSC) was established, dividing tumors into four classes. The GSC subtype definition did not rely on data clustering, but rather on sequential application of classifiers defined by other groups. Thus, the data set was first separated into basal and luminal through the application of the University of North Carolina (UNC) BASE47 classifier [38]. Then, the basal side was further subdivided by application of the UNC claudin-low classifier [39] resulting in a Claudin-low group and a non-claudin-low group termed only ‘basal’. The BASE47-luminal side was subdivided based on classification results of The Cancer Genome Atlas (TCGA) classifier [40], where TCGA class II cases formed the GSC ‘luminal-infiltrated’ subtype, while cases with TCGA clusters I, III, or IV formed the GSC ‘luminal’ subtype. While combining classifiers in this way could be rational for capturing important aspect of the two classifiers, it is notable that the survival endpoints were first analyzed for all classifiers both in the TCGA and in most patients (*n* = 200) from the NAC cohort. This analysis concludes: “In summary, the differences in OS by subtype and the apparent impact of NAC suggest that a classification into four subtypes would have the greatest clinical relevance” and in the next step the GSC is defined by combining classifications as described above. Consequently, the GSC is constructed to capture different clinical outcomes in the NAC and non-NAC cohorts. This approach is fundamentally different from all other classifiers that used only biological information to define the groups, and only subsequently tested subtypes for clinical associations. The pathologic response (defined as ypT < 2N0) in the NAC cohort was not significantly associated with molecular subtypes for any classification system tested when applied to the initial 200 cases. The MDA-luminal and MDA-p53-like categories responded in 63–64% of cases, whereas 46% among the MDA-basal subtype responded. Applying the Lund classifier (based on the first version of the Lund Taxonomy classifier [41]), the Lund ‘genomically unstable’ (GU) subtype had the highest response rate (63%) in the same 200 cases. For the newly developed GSC subtypes, the response rates (now for all 269 cases) were the following: GSC-luminal 47%, GSC-luminal-infiltrated 31%, GSC-basal 43%, and GSC-claudin-low 42%. It is unclear why the number with the response information was higher and the overall response rates were lower in the analysis based on GSC classification than for the other subtypes (Supplementary Tables S3 and S5 in Ref. [34]). Thus, the GSC-luminal, MDA-luminal/p53-like, and Lund-GU subtypes had the highest response rates, albeit not significantly different in the initial 200 patients in the cohort. As expected, and reported in the supplement of the original publication, the survival curve of major responders (ypT < 2 N0) was strongly improved compare to non-responders in the NAC cohort. Since pathologic response was associated with survival in the full cohort, it is notable that the improved survival outcome of the GSC-basal subtype in the NAC cohort was not accompanied by a higher pathological response rate. An explanation could be that the GSC subtypes were selected to capture survival differences between the NAC and reference cohorts, but not response differences within the NAC cohort.

In 2020, Taber and colleagues published an integrated multi-omics analysis that identified molecular correlates of cisplatin sensitivity in various clinical settings [20]. The cohort included transcriptomic analysis for 121 patients, of whom 44 received NAC and 77 received first-line cisplatin-based chemotherapy in the metastatic setting, albeit not population-based. Several genomic features were significantly linked to response, including the SBS5 mutation signature, a high number of indels, and the presence of *BRCA2* mutation. It also identified a significant difference in response rates between the consensus RNA-based molecular subtypes with basal-squamous (BASQ) (47%) having a lower response rate than the other subtypes (71%). Notably, no tumors were classified as luminal non-specified (LumNS), and the best responding consensus subtype was stroma rich with 14/18 patients responding. The source data from Taber et al. also allow us to tentatively overview if the DDR mutations and the molecular subtypes independently identify responders. As shown in Appendix A, the best responding subtypes (luminal unstable (LumU) and stroma rich) were not enriched for *ERCC2*, *BRCA2*, or DDR mutations. The highest fraction of responses potentially explained by these genomic predictors was actually seen in the poorly responding BASQ subtype, although the difference is small and not statistically significant.

Last year, Lotan and colleagues published a statistical meta-analysis of response and outcome for NAC-treated and untreated patients from four cohorts profiled with the Genomic Subtyping Classifier (GSC) [35], of which three were previously reported as original and full-length publications. The study included data on 247 stage cT2-4N0-3 patients treated with pre-operative chemotherapy and 354 stage cT1-4N0-3 patients who received RC without prior NAC treatment. The non-NAC arm included 31 clinically stage T1 tumors that were pT ≥ 2 after RC. Clinically T1 tumors are often of a luminal-like character. Since only the non-NAC cohort included clinically T1 tumors, it is not surprising that the NAC and non-NAC cohorts had an uneven distribution of GSC subtypes. The percentage of each GSC subtype that was given NAC was 35% for GSC-luminal, 45% for GSC-luminal-infiltrated, 40% for GSC-basal, and 54% for GSC-claudin-low. Furthermore, patients from one of the four cohorts had undergone circulating tumor cell (CTC) analysis, as part of a trial protocol where the CTC-positive patients were selectively recommended NAC, whereas CTC-negative patients were not [42]. The adherence to this recommendation was not included in the abstract, meaning that it is not known to what extent CTC-positive patients, unlikely to be cured by surgery alone [43,44], were selectively included in the NAC-cohort and CTC-negative patients in the non-NAC cohort. In the survival analysis, the NAC and non-NAC cohorts were compared regarding OS and cancer-specific survival (CSS) using a statistical weighting to adjust for age, sex, and clinical stage group (cT1–2 vs. cT3–4). The weighted cohorts were then directly compared with regards to outcome. The analysis showed no difference in CSS between the NAC and non-NAC groups within the GSC-luminal subtype, but a hazard ratio (HR) of 0.59 with 95% confidence interval (CI) 0.38–0.93 for NAC treatment within the other GSC-subtypes combined (luminal-infiltrated, basal, and claudin-low). Pathologic response data are not reported, and it is not specified if the data were not collected, if they exist and were not analyzed, or if they were analyzed but not included in the published article. The lack of response data, the selection of clinically T1 and of CTC-positive patients into specific arms of the study, and the fact that the main finding was no longer significant when excluding the 35 patients treated with adjuvant chemotherapy from the survival analysis, calls into question the generalizability of the results.

In the same year (2021), our group from Lund published a retrospective analysis of molecular classification using the MIBC-updated Lund Taxonomy (LundTax) and outcome after pre-operative chemotherapy and RC in a population-based Swedish cohort (*n* = 149) [36]. Pathologic response and survival were stratified by LundTax subtypes in the neoadjuvant subset of the cohort (*n* = 125) and compared to patients who received RC without peri-operative chemotherapy (*n* = 186) in a previously published consecutive cohort [30]. The results showed best response (52% pT0N0) for the genomically unstable (GU) subtype, compared to the basal/squamous (Ba/Sq) subtype (21% pT0N0). The third major LundTax subtype, urothelial-like (Uro), had an intermediate 31% pT0N0 rate with rates for UroA, UroB, and UroC subsets being 43%, 24%, and 25%, respectively. The findings were largely mirrored in the CSS data, where the GU subtype fared best in the neoadjuvant cohort HR 0.29, 95% CI 0.11–0.79, with the basal/squamous subtype as reference and adjusting for tumor stage. Again, the Uro subtype(s) were intermediate in terms of CSS. Notably, the UroC subset that represents Uro tumors that are molecularly most similar to the genomically unstable subtype also had better CSS compared to Ba/Sq (HR 0.37, 95% CI 0.14–0.94). The UroB subset, which are Uro tumors most similar to the Ba/Sq subtype, did not do significantly better than the Ba/Sq for either response (25% vs. 21% pT0N0 nor CSS (HR 0.7, 95% CI 0.26–1.9). Comparison to patients in the RC cohort who received no peri-operative chemotherapy revealed that pT0N0 was rare. While these rare pathologic responses occurred more often in Uro (10%) and GU (11%) than in Ba/Sq (4%), and the survival curves of both Uro and GU were located above that of Ba/Sq, the differences were much less pronounced. As has been previously described in the full Lund2017 cohort [30], this analysis of the MIBC subset without peri-operative chemotherapy again revealed no significant differences in survival outcomes between LundTax subtypes. The main drawback of this study is that it cannot rule out that some true prognostic difference between subtypes exists that was not identified with statistical certainty in the control cohort. Such prognostic differences could have contributed to the observed survival differences in the NAC cohort. The study also applied the Consensus classifier, as well as an extensive predefined IHC subtyping based on 13 immunostainings applied to tissue microarrays [45]. Although the concordance between LundTax RNA- and IHC-subtyping was not perfect (5 × 5 group concordance = 0.66), the response and survival stratified by IHC subtyping was similar to that obtained when stratifying by RNA subtypes. The Consensus classification also indicated that the LumU subtype (corresponding to the LundTax GU) had more responses compared to BASQ (53% vs. 25%,) and better stage-adjusted survival (HR 0.23, 95% CI 0.07–0.80). In addition to the LumU subtype, the stroma-rich consensus subtype had significantly better stage-adjusted survival than BASQ (HR 0.3, 95% CI 0.10–0.9), albeit without significantly higher response rates (37% vs. 25%, Chi-2 test *p* = 0.38).

In addition to these published studies, RNA-seq data exist for the SWOG S1314 trial in which patients received neoadjuvant MVAC or GC followed by RC [46]. Reports from this study focus on the pre-specified application of the COXEN method for predicting MVAC or GC sensitivity [47], which was not able to significantly stratify response in its respective treatment arms in 167 evaluable cases. Data on response stratified by molecular subtype have only been published in the form of an abstract, which indicated that, in 161 patients, no significant association with pathologic response was observed with the Consensus, TCGA, or MDA classifiers [37]. OS data for this trial will be analyzed, and stratified by COXEN score and molecular subtype, with additional follow-up.

### 3.2. Studies on NAC Response with IHC- or RNA-Panel Subtyping Data

Studies with IHC- or RNA-panel subtyping are summarized in Table 2. In 2015, Baras and colleagues published a study on molecular profiling and NAC response that made use of both RNA-based methods for discovery and IHC-based methods for validation [48]. Although this study did not intend to analyze molecular subtypes, the identified genes and proteins are associated with luminal- and basal-like molecular profiles, allowing subtypes to be inferred. Briefly, a gene expression cohort (*n* = 33) of NAC (GC)-treated patients was screened and 21 genes associated with response were identified. For six genes, antibodies against corresponding proteins were applied and the best separation of responders were achieved by combining *GDPD3* (high in responders) and *SPRED1* (high in non-responders). The direction for these markers was the same in the original RNA-data and in the IHC-validation (*n* = 37). To explore these results, we mapped 11 of the 21 genes to the Lund2017 RNA data set [45]. The analysis consistently showed that genes upregulated in the responsive cluster (including *GDPD3*) were highly expressed in luminal-like (Uro and GU) subtypes, whereas genes upregulated in the resistant cluster (including *SPRED1*) were highly expressed in the Ba/Sq subtype (Appendix A). The associations between these genes and subtypes were also confirmed in the TCGA data set [29] (data not shown). Since the mapped genes were associated with luminal and Ba/Sq profiles with a coherent directionality, one can assume that responses were enriched in the inferred luminal cases and depleted in inferred Ba/Sq cases in both the RNA and IHC cohorts of this study. The molecular classification tree proposed in the study identified none of four tumors with the most luminal-like profile (GDPD3+ SPRED1−) and six of six tumors with the most basal-like profile (GDPD3− SPRED1+) as NAC resistant. Most tumors, however, were double positive or double negative. In these cases, the negative status of the luminal-marker *GDPD3* was most strongly linked to resistance.

Zhang and colleagues presented an abstract and poster at the annual European Association of Urology congress in 2017 [49], but the data have not yet been published in a peer-reviewed journal, and the results should be taken as preliminary. In summary, 99 tumors from NAC-treated patients were subjected to IHC-based subtyping (KRT5/6, KRT14, GATA3) and the patient outcomes were compared to 97 patients not treated with NAC. No difference in the rate of pT0N0 was seen, but luminal-like cases had significantly more downstaging to pTa/pT1/pCISN0. Patients with a basal subtype had worse OS than those with a luminal subtype both with and without NAC treatment. After adjusting for clinical stage and adjuvant treatment, a significant survival benefit of NAC was found in the luminal (HR 0.46, 95% CI 0.27–0.78) but not in the basal subtype (HR 0.60 95% CI 0.29–1.20).

Recently, Font and colleagues published a retrospective analysis of IHC-based subtyping and chemotherapy response in 126 patients by applying established luminal and basal markers to TMAs [50]. The study cohort and methods are well described, with about 20% node-positive cases and 10% receiving carboplatin-based regimens. Tumors were classified as luminal if positive for luminal markers only (FOXA1, GATA3), basal if positive for basal markers only (KRT5/6/14), or mixed (FOXA1, GATA3, KRT5/6 positive, KRT14 negative). Interestingly, only two ‘double negative’ cases were identified, even though IHC-based classification usually identifies about 5–10% of MIBCs as double negative for these markers [24,45,55]. Possibly, the absence of such tumors in the cohort could be due to chance, or selection mechanisms could differ between studies and cause differential inclusion of, e.g., sarcomatoid, neuroendocrine, or other histologic variants that can display such a double-negative phenotype [56,57,58]. The 47 of 126 cases classified as BASQ-like had a higher rate of pathologic response (43%) than the luminal (27%) and mixed (20%) clusters. Notably, this study applied hierarchical clustering to arrive at the three-group classification. The basal cluster was the most stable of the three clusters, which is expected since the other two clusters share expression of both GATA3 and FOXA1. However, a closer inspection of the basal cluster shows that there are nonetheless subsets of cases with partial FOXA1/GATA3 expression and low KRT5 expression. Importantly, most of the responders within the BASQ-like group were seen among such atypical BASQ-like cases (see top and bottom parts of the BASQ cluster in Figure 2A in Ref. [50]). Possibly, different conclusions could be drawn from this cohort if the BASQ-like group were restricted to ‘core cases’ completely negative for FOXA1 and GATA3 and positive both for KRT5 and KRT14 according to the consensus definition of a basal-like IHC profile [59]. To exemplify further, if the cases in this cohort were rank ordered by ‘basalness’, i.e., a numeric score of KRT5+ KRT14+ GATA3− FOXA1−, the highest frequency of responders would not be among the most basal cases, but rather among those that were categorically classified as BASQ-like but with a relatively lower ‘basalness’.

During 2020 and 2021, four small studies applying two to four markers were published. Pichler and colleagues applied KRT5/6 and GATA3 to full section TURB block from 21 GC-treated patients [51]. The study found that 19/21 cases expressed GATA3 in over 25% of the cancer cells, and five of these 19 co-expressed KRT5/6. Two-thirds of the cohort were classified as responders (pT ≤ 1), including the two non-luminal cases (one potentially basal-like and one with a double-negative profile). The presence of double-positive cases is well documented in the literature (Consensus luminal-papillary (LumP) or LundTax UroB subtypes [60]) and should be expected. However, the low number of cases identified as non-luminal (GATA3−/low) could either have occurred by chance, or it suggests that full-section IHC with GATA3 may need calibration or subtype heterogeneity analysis to capture cases that would be basal-like by transcriptomic analysis. Taken together, the reported response rate in the cohort was high. Since most cases were classified as luminal-like or double positive, this would favor the luminal-like category. However, the two non-luminal cases also responded. The more stringent response criteria pT0N0 was not reported. Morselli and colleagues applied a four-marker IHC panel (KRT20, GATA3, KRT5/6, and CD44) to TURB tissue from 16 patients treated with NAC and radical cystectomy [52]. The results show that none of five cases with a clear basal-like profile (CK5/6+ and CD44+) had a pathologic complete response, whereas all five (of 11) classified as responders (pT ≤ 2) had a luminal-like profile. The only death caused by bladder cancer was from a basal-like tumor. Of the three complete responders (pT0N0), two expressed CK20 highly and two expressed KRT5/6 along the edge/margin, thus suggesting one case consistent with a GU subtype and two cases consistent with a Uro subtype [60]. Similarly, Jütte and colleagues applied another four-marker qPCR panel (KRT20, ESR1, ERBB2, KRT5) to TURB tissue from 54 patients treated with two to three cycles of GC followed by RC [53]. High expression of luminal markers (KRT20, ESR1, and ERBB2) and low expression of KRT5 was significantly associated with pathologic complete response (*p* = 0.009). Of 22 complete responders, 19 were luminal-like as defined by these markers. Finally, Razzaghdoust and colleagues applied a four-marker IHC-panel (KRT20, GATA3, KRT5, and KRT14) to TURB tissue from 63 patients treated with neoadjuvant gemcitabine and cisplatin or carboplatin followed by RC [54]. Of 16 responders (tumor free by post-treatment cystoscopy and CT scan), half were classified as basal (KRT5/6+ KRT20−) and this subtype had the highest response rate. Although 8/15 basal cases were responders, 4/12 luminal (KRT20+ KRT5/6−) cases also achieved a clinical response. In this study, double-negative and double-positive cases had the lower response rates (9% and 14%, respectively). Positivity of GATA3 and KRT14 were deemed to result in imbalanced groups with KRT14 being expressed in very few cases and GATA3 being expressed in almost all cases. As seen in the Appendix A, stratification by these markers resulted in 3/4 basal cases (GATA3− KRT5+) with a complete response compared to 5/19 with a GATA3+ KRT5− profile.

## 4. Discussion

We find that the reviewed scientific literature does not support the conclusion that the basal-molecular subtype of MIBC responds well to NAC. While many individual studies suggest a link between molecular subtypes and neoadjuvant treatment response, the directionality and effect sizes varied between studies. A potential explanation is that some studies identified a true association, whereas others obtained results in favor of a false conclusion. Another possibility is that most, or all, studies identified true associations which differed between patient populations. A third possibility is that all the significant associations described in the literature are false. Of the three potential explanations, we consider the first to be more likely, but the other two could not be ruled out. The reviewed research field consists nearly exclusively of retrospective cohort studies applying different methods to define and determine molecular subtypes. Furthermore, cohorts were seldom population-based and used different definitions of pathologic response. The evidence on the treatment predictive value of subtyping is uncertain because of a lack of studies employing a more optimal design to disentangle how molecular information can be translated into clinical decision-making in the setting of NAC.

In the retrospective cohorts that have been published so far, pathologic response is included in most studies as a primary endpoint to assess tumors’ sensitivity to systemic treatment. Although it is a surrogate endpoint for survival [61,62], it serves as a read-out of cancer cell death locally in the bladder close in time to the administration of treatment. It operates at the tumor level, unlike survival endpoints which operate at the patient level. This means that response is less susceptible to confounding by patient-level factors, such as age, comorbidity, or other inherent (known and unknown) host-factors that influence survival. However, pathologic response as outcome measure has limitations, as about 10% who receive no neoadjuvant treatment also achieve a pCR in the absence of neoadjuvant treatment [36,63], attributed to the diagnostic TURB. It has been estimated that 38% of pathologic responses can be attributed to TURB and thus are unrelated to the cancer’s intrinsic sensitivity to treatment [64]. Still, patients with a pCR have an OS risk ratio of 0.45 compared to patients without it [62]. The effect of pathologic response on survival also scales with the number of downstaging steps. This adds additional complexity since patients with a three-step downstaging from cT3-pTa can have better outcomes than those who are downstaged only one step (e.g., from cT2-pT1) [65]. This indicates that there is no threshold over which it is particularly important to respond, and that binary response criteria are arbitrary and give an undue influence of initial clinical stage over binary response evaluation. To exemplify, a greater response is required for a cT3 than for a cT2 tumor to cross a specific response threshold, but cT3 tumors are not necessarily shrinking less in response to treatment than cT2 tumors. In addition to response in the bladder, similar issues operate in parallel for nodal staging, which exacerbates the problem. In the studies reviewed here, several used different definitions of pCR (ypT0N0 [36], ypTa/pCIS [20], ypT < T1N0 [33], and ypT < T2N0 [34]), which is clearly problematic. Nonetheless, NAC similarly affects both the cancer burden in the bladder, which is measured by pathologic response, and the micrometastatic disease, which is what causes differences in patient outcome. Therefore, pathologic response is an important read-out of treatment effect despite being imprecise, noisy from the influence of TURB, and arbitrarily categorized.

Survival endpoints rely heavily on comparison with an untreated control cohort, without which it is not possible to separate a predictive from a prognostic survival difference. The ideal control cohort would be similar at baseline to the treated cohort, but this is usually not the case as patients receiving NAC are younger individuals with intact renal function compared to those who receive only cystectomy [66]. Thus, patients receive or do not receive chemotherapy for various reasons, which also affect baseline risk to varying degrees. Residual confounding is likely despite adjustment for tumor stage. Additionally, applying different inclusion criteria between treated and untreated cohorts, i.e., cT1 tumors or patients with nodal metastases that never would have been considered for NAC, also makes survival comparisons less reliable. Taken together, pathologic response and survival endpoints have different pros and cons, and an optimal experimental design allows both endpoints to be studied with as little risk for bias as possible.

The reviewed RNA-based subtyping studies on NAC and RC for MIBC did not show consistent results. The early study by Choi and colleagues pointed toward worse response for the stroma-infiltrated p53-like subtype compared to luminal or basal-like subtypes, but there was no survival difference [32]. Similar results were obtained by the same group when investigating the same subtype-scheme in a phase II trial on dose-dense MVAC and bevacizumab in 38 patients, where the p53-like subset did poorly in terms of survival without any significant difference in response [33]. These two studies make a case for poor response and outcome of the p53-like subset, but they do not suggest any robust relative benefit of either the basal or the luminal subtype.

On the other hand, the study by Seiler and colleagues reports that the basal subtype had unexpectedly good survival outcome compared to controls, mainly the TCGA cohort [34]. However, there are several points of concern that warrant caution. As described in the results section, the GSC subtypes were to some extent defined in this same cohort, with the aim to detect survival differences captured by other classifiers. Additionally, the OS difference was not mirrored by a similar difference in pathologic response. In fact, response did not even trend in the same direction, and instead the highest (not significant) response rate was seen in the luminal subtype regardless of classification system. Second, the survival comparison rests heavily on the control cohort, which was a historic control (i.e., not generated in the study from the same contributing centers, but a completely external data set). If the basal subtype by chance had a poor outcome in the control cohort (if, e.g., patients with basal tumors in the control cohort had metastatic disease more often), one would expect exactly the observed result. Survival but not response would then appear to have improved in the NAC cohort. The prognostic value of subtyping in MIBC cohorts depends on the span and variation of clinical risk factors like the clinical TNM stage. In cohorts with a large span on clinical risk, molecular subtypes are more prognostic. In a NAC cohort, the eligibility for treatment always somewhat constrains the variation in clinical risk. In the TCGA cohort, this variation is large and consequently molecular subtypes are indeed highly prognostic [29], as opposed to population-based and more stringently selected cohorts [30,31]. Thus, baseline differences between the NAC and the control cohorts might affect the survival analysis, and together with a lack of confirmatory pathologic response data and CSS as an outcome measure, generalizability of the reported results await confirmatory studies. The later study by Lotan and colleagues also applied the GSC subtypes, although with heterogeneous study inclusion. While this study adjusted for some variables, it selectively included clinical T1 tumors in the untreated arm and cases with known disseminated disease in the treated arm. Unfortunately, the authors did not report pathologic response information, which further urges caution given that they examined a commercially available biomarker (Decipher^®^) [35]. Finally, even if these reservations are ignored, the statistical significance of the main finding falls when patients treated with adjuvant chemotherapy are excluded from analysis, something that arguably should have been done from the beginning.

Two studies with relatively similar results are those by Taber et al. and Sjödahl et al., which analyzed Danish and Swedish patients, respectively [20,36]. Both studies show that response and survival was poor in the basal/squamous subtype by LundTax and Consensus classifiers. The Taber study is based on both NAC- and cisplatin-based chemotherapy in the first line setting for metastatic bladder cancer [20]. It has been pointed out that the findings in these studies are not in direct contradiction with that of Seiler and colleagues [34] since the GSC-basal subtype, despite the similar name, does not show a big overlap with the basal/squamous subtype by other systems [67]. Even if the results are not incompatible with that of Seiler et al., these studies go further by showing that luminal-like subtypes had both better response and survival than Ba/Sq. Interestingly, both studies also identify the stroma-rich consensus subtype to have particularly good outcomes, which contrasts with earlier described results with the MDA p53-like subtype. This motivates further study on the role of stromal signatures and NAC response to resolve the discrepant results with the conceptually similar MDA-p53-like and Consensus stroma-rich subtypes in these cohorts. Immune and stromal content contribute to the classification with most transcriptomic classifiers, which makes it difficult to assess the role of such signatures within molecular subtypes. The LundTax classifier differs from all others in that it strives to assign subtypes based on the expression patterns of cancer cells, reducing the influence of immune and stromal content on classification [68]. Scores for immune or stromal infiltration can then easily be analyzed independently in addition to molecular subtyping. Being designed to primarily capture the phenotype of the cancer cells, the LundTax classification could therefore be a good framework to study subtype-dependent effects of signals from the tumor microenvironment.

Finally, the SWOG S1314 trial showed no significant results with any of three different classifiers (MDA, TCGA, Consensus). The results have only been presented in the form of an abstract, which states that with additional follow-up the associations with survival will be analyzed [37].

Altogether, results obtained with RNA-based subtyping studies on NAC response can be grouped into three types. First, analyses with MDA-classification in several small cohorts show that p53-like tumors may have either poor response or poor survival. There was no consistent difference in relative response rates of luminal-like versus basal-like subtypes in these cohorts [32,33]. Second, analyses with the GSC classifier suggested better survival for the GSC-basal and an absence of survival benefit for the GSC-luminal subtypes. As with the SWOG S1314 trial, neither of the GSC subtyping studies detected a difference in response [34,35]. Third, the Taber et al. and Sjödahl et al. studies both showed that luminal-like subtypes (i.e., LundTax Uro and GU) and the Consensus stroma-rich subtype had better response rates and survival after NAC, compared to the poorly responding Ba/Sq subtype [20,36]. In the population-based Sjödahl study, the survival advantage for the GU subtype was significant after adjusting for clinical stage and was not significantly present in a control cohort without peri-operative chemotherapy treatment from the same centers favoring a predictive rather than prognostic effect in this cohort. Interestingly, the finding that luminal-like subtypes are more responsive than basal are in line with what is known on the response of histologic variants of MIBC. Micropapillary tumors are luminal-like [69] and have been described to have a good response to NAC [70,71], whereas squamous-cell carcinoma of the bladder has a poor response to NAC [72,73,74,75]. One interpretation of histologic variants is that they represent extreme forms of the various differentiation states of urothelial carcinoma captured by molecular subtypes [56]. To reconcile this view with a NAC-responsive basal subtype and a NAC-resistant luminal subtype would require that the directionality of the effect on response changes completely from luminal subtype to micropapillary variant and from basal subtype to squamous-cell carcinoma.

Results from studies that use IHC- or RNA-panel-based subtyping are even more heterogeneous than for RNA-based classification. The studies by Font et al. and Razzaghdoust et al. show better responses for the basal subtype, but the survival data do not provide statistically robust support for this conclusion in either study [50,54]. More studies support a good response in luminal-like subtype(s), but results are generally weak and the luminal subtype had to be inferred from markers that are not validated surrogates for subtyping. The two exceptions are the studies by Sjödahl et al. in which the IHC subtyping was pre-defined with clear results on both response and survival [36], and the study by Jütte et al., which found a statistically quite robust association between the luminal-like category and response [53]. Still, it is not surprising that more heterogeneous results are found with panel-based subtyping than with RNA-based studies. Each study has an unknown classification error since surrogate panels only use a limited number of markers. Furthermore, intra-tumor heterogeneity may affect studies differently depending on the platforms and markers that are used for subtyping. Future studies should consider including measures of biological heterogeneity to decrease the risk of a false negative result. Despite these caveats, if similar samples of the same population are studied, one would expect classification error to average out and most study results to point in the same direction. Instead, it appears that there are results pointing in all possible directions. A potential explanation could be that analytic flexibility influences panel-based tumor classification more than when using transcriptomic classifiers which are often pre-defined. Studies may thus have tended to report the markers, cut-offs, and marker-panel combinations that led to a statistically significant result in the desired direction.

## 5. Limitations

The 14 articles included in this review differed substantially in several critical aspects of study design, including selection of study population, sample size, chemotherapy regimen, RNA- or panel-based classification method, response criteria, and clinical endpoints. These sources of variability in study design are summarized in Table 3 and represent major challenges for the field going forward.

## 6. Conclusions

Going forward, several studies show an indication that subtypes may be predictive of NAC response. To avoid the negotiable limitations discussed above, the field must now strive to produce studies of better quality that test pre-specified hypotheses in prospective designs (ISRCTN15459149). Another important aspect to mitigate bias in this field is that any new study, regardless of design, should report both response and survival outcomes case by case and make all molecular data publicly available. In addition to any selected subtyping method, the standard molecular subtyping methods (Consensus classification for RNA-based classification and the basal-luminal panel KRT5/KRT14/GATA3/FOXA1 for IHC-classification) should also be applied. While subtypes can be grouped to increase statistical power, grouping very different tumors into one category without testing if and how heterogeneity affects the results is problematic. With these considerations, it should be possible to perform translation of molecular classification for MIBC patients receiving NAC into the clinic within the near future. After all, to refrain from NAC based on predictive information still leaves the door open for adjuvant chemotherapy after RC [76].

## Figures and Tables

**Table 1 cancers-14-01692-t001:** Studies on NAC response with original RNA subtyping data.

Study	*n*, NAC + RNA Data	Regimen	k Subtypes, Classifier	Response Criteria	Response	Survival Post-NAC
Choi et al. [32]	100	MVAC	3, MDA	pT0 or pT1 ^1^	↑Lum, Basal ↓p53-like	N.S. ↑Lum ↓p53-like
McConkey et al. [33]	38	4 MVAC + Bev	3, MDA	<pT2	N.S. ↓p53-like	↑Basal ↓p53-like
Seiler et al. [34]	251	≥3 MVAC GC	4, GSC	pT < 2 N0	N.S. ↑Lum	↑Basal, Lum ↓Lum-inf, Claudin-low
Taber et al. [20]	44	GC	6, Consensus	≤pTa, cis,N0	↑Stroma-rich ↓Ba/Sq	↓Ba/Sq
Lotan et al. [35]	247 ^2^	Cisplatin-based	4, GSC	Not reported	Not reported	↓NAC benefit Lum
Sjödahl et al. [36]	125	≥2 MVAC GC	7, LundTax	pT0N0	↑GU ↓Ba/Sq	↑Lum (GU, UroC) ↓Ba/Sq
Lerner et al. abstract [37]	161	4 MVAC GC	3, Consensus; 3, TCGA; 3, MDA	pT0	N.S.	Not reported

^1^ pT1 was counted as a response only for patients who fulfilled certain high-risk criteria. ^2^ 82 of the NAC-treated patients overlapped with the study by Seiler et al. Abbreviations: NAC, neoadjuvant chemotherapy; MVAC, metotrexate vinblastine adriamycin and cisplatin; Bev, bevacizumab; GC, gemcitabine and cisplatin; MDA, MD-Anderson; GSC, genomic subtype classifier; LundTax, Lund taxonomy; TCGA, the cancer genome atlas; pT, pathological T-stage; cis, carcinoma in situ; Lum, Luminal; Ba/Sq, basal/squamous; GU, genomically unstable; N.S., not significant; UroC, urothelial-like C.

**Table 2 cancers-14-01692-t002:** Studies on NAC response with IHC- or RNA-panel subtyping.

Study	*n*, NAC	Regimen	k, Subtypes and Classifier	Response Criteria	Response	Survival Post-NAC
Baras et al. [48]	33 + 37	≥2 GC	2, (inferred Lum, Ba/Sq)	Tumor reduction, ≤pT1	↑Lum, ↓Ba/Sq	Not reported
Zhang et al. abstract [49]	99	≥2 GC	2, (Lum, Basal)	pCR pT0N0, pPR < pT2N0	pCR N.S. pPR ↑Lum ↓Basal	↑NAC benefit Lum
Font et al. [50]	126	GC CMV GCa	3, (Lum, BASQ, mixed)	pT0N0	↑BASQ ↓Lum ↓Mixed	N.S.
Sjödahl et al. [36]	125	≥2 MVAC GC	5, LundTax	pT0N0	↑GU ↓Ba/Sq	↑GU, Uro ↓Ba/Sq
Pichler et al. [51]	21	GC	2, (inferred Lum, Basal)	≤pT1N0	pR in 2 basal/DN, and in 57% of Lum	Not reported
Morselli et al. [52]	16	3 GC	2, (inferred Lum, Basal)	≤pT1N0	All 5 pR in Lum. 0/5 in Basal	N.S.
Jütte et al. [53]	54	2–3 GC	2, (inferred Lum, Basal)	pT0N0	↑Lum	Not reported
Razzaghdoust et al. [54]	63	GC GCa	4, (Lum, Basal, DN, DP)	Clinical (cystoscopy, CT)	↑Basal ↓DP, DN.	N.S.

Abbreviations: NAC, neoadjuvant chemotherapy; GC, gemcitabine and cisplatin; CMV, cisplatin metotrexate and vinblastine; GCa, gemcitabline and carboplatin; MVAC, metotrexate vinblastine adriamycin and cisplatin; Lum, luminal; Ba/Sq, basal/squamous; BASQ, basal squamous-like; LundTax, Lund taxonomy; DN, double negative; DP, double positive; pT, pathologic T-stage; pCR, Pathologic complete response; pPR, pathologic partial response; CT, computerized tomography; N.S., not significant; GU, genomically unstable; pR, pathologic response.

**Table 3 cancers-14-01692-t003:** Sources of variability in experimental design between the 14 included studies.

Study population:	All studies included cT2-4a. 7 studies included N+ disease
Sample size (*n*, NAC):	Range: 16–251. Median: 85
Chemotherapy regimen:	12 studies included GC. 6 included MVAC. 6 only GC. 2 only MVAC
RNA-based subtyping:	5 RNA classifiers, 3–7 subtypes
Panel-based subtyping:	4 schemes, 2–5 subtypes. 63% only 2-tier (Lum and Basal)
Response criteria:	5 studies used pT0N0. 7 studies used <pT2. 2 studies did not report pR.
Clinical endpoints:	13 studies reported response. 9 reported survival.

Abbreviations: NAC, neoadjuvant chemotherapy; GC, gemcitabine and cisplatin; MVAC, metotrexate vinblastine adriamycin and cisplatin; Lum, luminal; pT, pathologic T-stage; pR, pathologic response.

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
