# Peer review of "Molecular Subtypes as a Basis for Stratified Use of Neoadjuvant Chemotherapy for Muscle-Invasive Bladder Cancer—A Narrative Review"

_cancers, 2022, doi:10.3390/cancers14071692_

Round 1

Reviewer 1 Report

This is a timely, comprehensive, and well-written review.  The authors are leaders in the field.  This will be an important addition to the literature.  I have only the following minor suggestions:

A few typos: 

  • page 2, line 86 has an extra "that"
  • page 3, line 117, delete "confers"

Although not the topic of this review, authors could acknowledge in the Introduction that bladder-sparing chemoradiotherapy is an alternative to NAC followed by RC and is supported by multiple international guidelines as a primary curative option for select patients.

Tables 1 and 2 are important and useful.  In the final published version, increasing the spacing might make them more easily readable.

Although I agree with their sentiment, the second sentence of the Discussion (page 10, lines 436-439) seem somewhat contradictory....ie, "an association exists, but we don't know which direction."  A critic might say that if there is no sense of the direction, then its equally (or more) plausible that an association does NOT exist.  It would be worth clarifying this language to avoid an apparent contradiction.

The authors discuss several important considerations/caveats in the Discussion.  It may be worth creating a Table or Figure that summarizes these potential confounding issues (such as differences in populations and endpoints across studies, differences in molecular subtyping schemes, etc).

One issue that is discussed in this field but is not directly addressed in the Review is that of intra-tumoral heterogeneity.  It is plausible that analysis of multiple samples from the same patient at the same point in time may reveal different subtypes due to intra-tumoral heterogeneity.  It would be useful to discuss this issue in principle as well as any relevant data.

Although I realize that its not the primary topic, it would also be interesting to discuss NAC-induced changes in subtype that have been characterized by Seiler and others (ie, pre- vs post-NAC subtypes from the same patient).

Reviewer 2 Report

The authors conducted a thorough and comprehensive review of chemotherapy in different molecular subtypes of MIBC. The paper is extensive and detailed. It highlights most of the major issues regarding the topic. There are number of minor comments which the authors are encouraged to carefully consider:

  • The authors provided a methods section, but we don’t feel that narrative reviews are necessary. Despite the description of the included articles being detailed, we would tentatively recommend the authors expand the data on the papers excluded – did you intend to include only the major works on the topic? Did you include only recent publications? Did the authors aim to include most of the papers which they were able to obtain?
  • The authors are also encouraged to add a detailed limitations section at the end of the manuscript. Most of the limitations are already mentioned in the text, however, consider highlighting the major gaps in our knowledge in this section. This may pave the way for further research on this particular topic.
